# Research Progress and Prospects of Nanozyme-Based Glucose Biofuel Cells

**DOI:** 10.3390/nano11082116

**Published:** 2021-08-19

**Authors:** Phan Gia Le, Moon Il Kim

**Affiliations:** Department of BioNano Technology, Gachon University, Seongnam 13120, Korea; giaphan2008@gmail.com

**Keywords:** glucose biofuel cell, nanozymatic biofuel cell, enzymatic biofuel cell, enzyme mimic, electron transfer

## Abstract

The appearance and evolution of biofuel cells can be categorized into three groups: microbial biofuel cells (MBFCs), enzymatic biofuel cells (EBFCs), and enzyme-like nanomaterial (nanozyme)-based biofuel cells (NBFCs). MBFCs can produce electricity from waste; however, they have significantly low power output as well as difficulty in controlling electron transfer and microbial growth. EBFCs are more productive in generating electricity with the assistance of natural enzymes, but their vulnerability under diverse environmental conditions has critically hindered practical applications. In contrast, because of the intrinsic advantages of nanozymes, such as high stability and robustness even in harsh conditions, low synthesis cost through facile scale-up, and tunable catalytic activity, NBFCs have attracted attention, particularly for developing wearable and implantable devices to generate electricity from glucose in the physiological fluids of plants, animals, and humans. In this review, recent studies on NBFCs, including the synthetic strategies and catalytic activities of metal and metal oxide-based nanozymes, the mechanism of electricity generation from glucose, and representative studies are reviewed and discussed. Current challenges and prospects for the utilization of nanozymes in glucose biofuel cells are also discussed.

## 1. Introduction

Currently, humans are facing a shortage of energy sources and a wide range of environmental challenges caused by over-exploitation and over-consumption of fossil fuels. To overcome this crisis, new energy sources with sustainable and environmentally friendly characteristics are urgently required [1,2,3,4,5]. In the field of bio-electrochemical research, biofuel cells have emerged as an alternative energy conversion device to generate electricity from biomass while simultaneously preventing environmental pollution. Moreover, the biofuel cell is recognized as a key factor for realizing self-powered sensors, wearable devices, and implantable devices [6]. Biofuel cells can be categorized into three groups: microbial biofuel cells (MBFCs), enzymatic biofuel cells (EBFCs), and enzyme-like nanomaterial (nanozyme)-based (nanozymatic) biofuel cells (NBFCs) (Figure 1) [4,6,7].

The MBFC is a system that utilizes a microorganism as a reactor to catalyze the oxidation of biomass or waste and thus convert biochemical energy to electrical energy [1]. Fundamentally, the prototype architecture of MBFCs includes the anode, cathode, and membrane between the two (Figure 1A). The anodic compartment consists of electrochemically active microorganisms on a supportive electrode, and exoelectrogens, as defined by Logan et al. [8], play a role in donating electrons to the electrode by oxidizing substrates, whereas the large amount of oxygen in the cathodic compartment can accept electrons and protons to form harmless water. Supportive electrodes can be fabricated from carbon-based materials such as carbon fibers, carbon sheets, carbon cloth, carbon graphite, and carbon nanotubes [9,10,11]. Membrane is also an important component, That inhibits the diffusion of mediators and substrates, thereby reducing unwanted flux between the two electrodes but keeping them ionically and chemically conjugated [1]. Without membrane, severe biofouling can happen on the electrode surface, yielding instability and low efficiency in power generation during long-term operation. MBFCs have diverse biofuels, including wastewater, marine sediment soil, freshwater soil, and active sludge [1,12,13]. Electron transfer in MBFCs can be performed through direct contact between bacterial pili and electrode or short-range electron transfer [14], and indirect processes are possible via electron shuttles [15]. Although the MBFC possesses many advantages in waste processing and environmental protection, the critical drawbacks that limit its wide application and commercialization are significantly low power output and extreme difficulty of electron transfer control inside the microorganisms.

Unlike MBFCs, EBFCs catalyze the oxidation of biofuels with the assistance of natural enzymes to produce electricity. Biofuels for EBFCs are generally sugar relatives, such as glucose, sucrose, fructose, alcohols, including ethanol and methanol, organic acids, and organic salts such as sulfite salts [4,16,17]. Among them, the glucose-based EBFC is a well-known object because of the ubiquitous nature of glucose in the physiological fluids of plants, animals, and humans with high potential energy, facile mass production, low financial expenditure, and biocompatibility [18]. The estimated availability of glucose concentration in the human blood is approximately 2–10 mM [19], which is sufficient for developing enzymatic glucose biofuel cells. Like MBFCs, the EBFC comprises an anode and cathode, but the membrane may not be involved because of the high substrate specificity of natural enzymes on each electrode (Figure 1B) [20]. In the glucose biofuel cell, the anode generally consists of two kinds of enzymes, glucose oxidase (GOx) and catalase, to convert glucose to gluconolactone with hydrogen peroxide (H_2_O_2_) and H_2_O_2_ to water with oxygen, respectively. The cathode comprises laccase, which catalyzes the conversion of oxygen into harmless water. During the catalysis of one glucose molecule, two electrons are transferred through the electrode either directly or indirectly, which is also important for cell performance [4,5]. Glucose-based EBFCs have attracted much attention because of their capability to utilize glucose in physiological fluids to generate electricity, as well as improved power output in comparison with MBFCs. Nevertheless, glucose-based EBFCs have many limitations derived from natural enzymes, such as easy denaturation and instability, high production cost, and difficult electron transfer. Thus, exploring an alternative to natural enzymes that resolve these limitations is urgently needed. [7].

In this scenario, nanozymes take the spotlight with their preferable intrinsic peculiarities in comparison to those of natural enzymes, such as long-term stability, ease of synthesis with low cost, and tunable enzyme-mimicking activities, that present them as potential catalytic material for developing a glucose biofuel cell, namely the nanozyme-based glucose biofuel cell (Figure 1C). Nanozymes are functional nanomaterials having an ability to mimic the actions of natural enzymes, and until now, nanozymes typically include metals, metal oxides, metal chalcogenides, nanocarbons, and their composites, that induce distinct catalytic functions [21]. Nanozymes have been exploited in a wide range of applications for biosensors, environmental treatments, therapeutics, and particularly for glucose biofuel cell. To the best of our knowledge, no review paper has specifically discussed nanozyme-based glucose biofuel cell. In this review, we describe recent research progress on the representative synthetic strategies and catalytic characteristics of nanozymes, mechanisms of electricity generation from glucose, and application studies. We also describe the current challenges and prospects of advanced glucose-based NBFCs, based on the unique properties of nanozymes.

## 2. Synthetic Strategies and Catalytic Characteristics of Nanozymes to Replace Natural Enzymes in Glucose Biofuel Cells

To develop glucose-based NBFCs, nanozymes that mimic GOx and catalase are required to construct an anode for catalyzing glucose oxidation without accumulating H_2_O_2_, as well as laccase-mimicking nanozymes to construct a cathode for accepting electrons, which are produced and transferred to the cathode during glucose oxidation. Glucose-based NBFCs can be constructed by placing appropriate nanozymes in either both electrodes or a single electrode. Diverse types of nanozymes have been reported to mimic GOx, catalase, and laccase, and they can be categorized into noble metal and metal oxide-based nanozymes based on their composition. In this section, we describe the representative synthetic strategies and catalytic activities that are essentially utilized to develop glucose biofuel cells. 

### 2.1. Synthetic Strategies of Nanozymes

Nanozymes are generally synthesized via physicochemical routes similar to conventional nanomaterials. Different methods are available, and depending on the experimental orientation and application purpose, researchers can select between the appropriate top-down or bottom-up approach. The top-down approach includes a solid-state reaction route, in which the starting materials are scaled down to the synthesized product via ball milling, nanolithography, sputtering, and thermal decomposition processes; the bottom-up approach is a wet chemical route comprising sol-gel, reverse micelle, chemical vapor decomposition, pyrolysis, biosynthesis, microwave-assisted, and flow synthesis processes [22]. Between the two, the bottom-up approach is considered more efficient, and thus, it has been generally employed to synthesize diverse kinds of nanozymes. The bottom-up approach is capable of precisely controlling the size, morphology, crystalline structure, and surface properties of nanostructures. These features not only affect the physicochemical properties, but also significantly affect enzyme-mimicking activities. For example, smaller nanozymes around 10 nm in their diameter generally yielded higher activity than larger nanozymes, possibly due to their higher surface-to-volume ratio to combine with substrate [23]. Nanozymes preserving more active crystallographic facets showed higher activity due to higher surface reactivity [24]. Moreover, recent studies have been conducted to chemically mimic the structures of natural enzymes, including the active center or substrate-binding pocket, that yielded further enhancement of activity as well as selectivity toward the target substrate [25]. Therefore, material processing is a key factor for tailoring an arbitrary nanomaterial to obtain the required properties and realize the desired applications. 

### 2.2. Enzyme-Mimicking Characteristics of Nanozymes

Natural enzymes are categorized into six groups based on the type of catalytic reaction: oxidoreductase, transferase, hydrolase, lyase, ligase, and isomerase [26]. Four groups have already been mimicked by nanozymes, including oxidoreductase [27,28], hydrolase [29,30], isomerase [31,32,33], and lyase [34]. Diverse material types, such as noble metals, bi- or tri-metallic alloys, metal oxides, carbon, and hybrid-like metal-organic frameworks, have been reported as nanozymes. As nanozymes mimicking GOx, catalase, and laccase are utilized to develop glucose biofuel cells; the catalytic features of these nanozymes are discussed with recent examples. 

#### 2.2.1. Nanozymes with GOx-like Activity

GOx plays a critical role in generating electrons by catalyzing the oxidation of glucose within the biofuel cell. Complete oxidation of one glucose molecule can theoretically produce up to 24 electrons; however, this process involves a series of catalytic reactions. For the GOx-mediated oxidation of one glucose molecule, two electrons are formed by the production of one gluconolactone and one H_2_O_2_. Based on the high research and application significance, diverse types of nanomaterials comprising noble metals and metal oxides have been studied as GOx-mimicking nanozymes [35,36,37,38,39]. Among them, gold nanoparticles (Au NPs) have been a focus of investigation, and interestingly, their glucose oxidation mechanism was proposed to be the same as that of natural GOx [35], which is highly affected by morphological features, including size, and reaction environments, including temperature, pH, and reaction time [37] (Figure 2). Interestingly, the GOx-like activity of Au NPs was further improved by utilizing a molecularly imprinted polymer functionalized on their surface, resulting in a 270-fold higher catalytic activity than that of bare Au NPs [39]. In addition to Au NPs, other nanomaterials composed of metals, including Pt, Pd, Ru, Rh, and Ir, or metal oxides, including MnO_2_ and CeO_2_, have been reported to serve as GOx-like nanozymes, which have high potential for use in biofuel cell systems [40,41]. 

GOx-like nanozymes developed to date have been demonstrated to be utilized for catalyzing the oxidation of glucose on the anode to generate electrons for glucose biofuel cells. However, there are many unresolved problems limiting their practical applications, such as relatively low catalytic activity, poor substrate specificity, and limited electron transfer. The development of advanced GOx-like nanozymes has great significance in glucose-based NBFCs and other applications utilizing glucose. 

#### 2.2.2. Nanozymes with Catalase-like Activity

Natural GOx or GOx-like nanozymes catalyze glucose oxidation and produce H_2_O_2_ as a by-product, which has a detrimental effect on the electrode, resulting in decreased activity, lowered power output, and shortened lifetime. To overcome this obstacle, catalase, which converts H_2_O_2_ into O_2_ and H_2_O, and therefore preventing undesirable phenomena, needs to be involved in the construction of the anode of the glucose biofuel cell. Specifically, the enzymatic removal of H_2_O_2_ prevents the deleterious effects for GOx, as well as increases current density at the anode by its electrochemical reduction [5]. CeO_2_ nanoparticles are typical examples of catalase-like nanozymes [42,43]. Pirmohamed et al. [42] reported an important finding that the Ce^3+^/Ce^4+^ ratio determines the types of enzyme-mimicking activity of CeO_2_ nanoparticles. Catalase activity was predominant when the Ce^3+^/Ce^4+^ ratio was low. The mechanism for the catalytic behavior of CeO_2_ nanoparticles was explained by the electron exchange between the Ce^3+^ and Ce^4+^ present on the nanoparticles, arising from the H_2_O_2_ absorption in the oxygen vacancies of the CeO_2_ crystalline surface and the concomitant production of H_2_O and O_2_ by the electron-transfer reaction [43]. Once H_2_O and O_2_ were generated and detached from the surface, another H_2_O_2_ would subsequently attach to the CeO_2_ surface and continue the catalysis chain. Coordination polymer-based nanozymes were shown to exhibit catalase-like activity [44]; Fe^3+^ and adenosine monophosphate-coordinated nanoparticles exhibited catalase-like activity at neutral pH, while peroxidase-like activity was observed at acidic pH values via Fenton chemistry. 

Catalase-like nanozymes cannot be utilized solely for glucose biofuel cell development; however, bi-enzymatic nanozymes showing GOx and catalase have been reported to serve as an anode in glucose biofuel cells [45,46]. In this regard, finding efficient catalase-like nanozymes or bi-enzymatic nanozymes showing both GOx and catalase-like activity, for preparing an efficient anode in glucose biofuel cells is crucial for decomposing the byproduct H_2_O_2_ and enhancing the power output and extending the lifetime of glucose-based NBFCs. 

#### 2.2.3. Nanozymes with Laccase-like Activity

Laccase catalyzes the reduction of O_2_ to H_2_O by accepting electrons. The active site in natural laccase has four coppers, which are classified into three types: T_1_, T_2_, and T_3_. The substrate is oxidized at the T_1_ Cu site, and then transferred to the T_2_/T_3_ trinuclear Cu site, where oxygen is reduced to H_2_O [47,48]. Inspired by the active site structure, recent laccase-mimicking nanozymes have been designed and synthesized based on their molecular shapes. An interesting example is the imitation of the active site of natural laccase via the coordination of Cu^+^/Cu^2+^ with histidine and cysteine as ligands to form a laccase-like nanozyme (Figure 3) [47]. The laccase-like activity of the nanozyme was so efficient to yield even a four-order higher catalytic efficiency than that of natural laccase, presumably due to the presence of more active sites than natural laccase. Similarly, nanozymes having the active site-mimicking structures coordinated with Cu with cysteine-aspartic acid dipeptide or DNA-copper hybrid nanoflowers were also reported to exhibit laccase-like activity, which was higher and more stable than that of the natural counterpart [48,49]. 

Within the biofuel cell system, laccase can be utilized for developing cathodes, and many metal-based catalysts, including Pt, Mn, Fe, Co, and Ni, are generally efficient for preparing the cathode [50,51,52,53]. When we choose laccase-like nanozymes on cathode, general criterion is to maximize open circuit voltage (OCV) by considering the employed nanozymes on the anode, that can yield the highest power output. For their widespread utilization, unresolved problems of metal-based catalysts, such as high production costs, toxicity to humans and ecosystems, and limited substrate specificity, need to be addressed. Particularly, for the construction of miniaturized biofuel cell, the specificity toward target substrate enables simple assembly of both the anode and cathode without the need for membrane, thus reducing biofuel cell volume significantly. Moreover, unwanted side reactions can be effectively avoided, which is also important to safely generate electricity from the physiological molecules like glucose. Thus, the exploration of laccase-like nanozymes exhibiting superior catalytic performance with high conductivity is important for future applications of glucose biofuel cells.

## 3. Recent Research Examples of Glucose-Based NBFCs

The glucose-based NBFC was composed of an anode, a cathode, and an electrolyte containing glucose. Depending on the working environment, components in the NBFC can be modified to improve power output, conductivity, OCV, stability, and lifetime. In this section, we briefly summarize recently reported and representative examples of glucose-based NBFCs. We categorize our discussion into two main groups based on the types of nanozymes employed, primarily noble metals and several metal oxide-based ones. In most studies, these nanozymes were anchored on carbon-based materials, and then utilized to construct electrodes of the biofuel cell system, which could provide much enhancement in electrical conductivity as well as other catalytic advantages including facilitated substrate or mediator transfer. 

### 3.1. Noble Metal-Based Nanozymes for Glucose-Based NBFC

Noble metals such as Au, Pt, and Pd, which show outstanding GOx-mimicking activity, have been widely applied in glucose biofuel cells [54,55,56]. Xie et al. [54] introduced an Au nanozyme-based glucose biofuel cell, where Au salt was dispersed into a glucose electrolyte and deposited on the electrode during glucose oxidation to obtain a high-performance NBFC. Another NBFC comprising GOx-like Au nanowires as the anode and Pt/carbon as the cathode was reported [55]. Thanks to the large surface area and small particle size of the Au nanowires, a high-power output of 126 µW/cm^2^ and an OCV of 0.425 V were obtained. Another glucose-based NBFC incorporating an Au NP-based anode, graphene-based cathode, and Nafion membrane was developed to yield a high-power output of 10.7 mW/cm^2^, which is among the best results for glucose-based NBFCs [56]. 

Binary [50,57,58,59,60,61,62] or ternary [58,62] noble metal alloys have also been employed for developing glucose-based NBFCs, based on their synergistically enhanced enzyme-like activity and ability to circumvent the poisonous intermediates generated during the catalytic oxidation at the anodic electrode [51,58,60,61]. Chu et al. [61] developed a Pt/Au alloy-based glucose biofuel cell consisting of an anode of a GOx-like Pt/Au nanozyme and a cathode of graphene with a Nafion membrane. In this study, the optimized Pt/Au ratio (1:4) and glucose (6 mM) with saturated oxygen environment yielded good biofuel cell performances with power output of 0.32 mW/cm^2^ and OCV of 0.42 V. Guo et al. [50] designed and prepared another glucose-based NBFC consisting of nanoporous Au-PtBi as the anode, Pt/carbon as the cathode, and glucose electrolyte mixed with NaOH. At an optimized glucose concentration (0.5 M), a high-power output performance of 8 mW/cm^2^ and an OCV of 0.95 V were achieved. A bimetallic nanozyme-based glucose biofuel cell comprising an anode constructed of GOx-like Au_80_Pt_20_/carbon and a cathode of Pt/carbon with a glucose solution containing KOH, has been reported [51]. In this study, a series of experimental parameters were tested, and the glucose and KOH concentration, reaction temperature, and flow rate of glucose solution were optimized to be 0.6 M, 4 M, 328 K, and 50 mL/min, respectively. By adopting the optimized conditions, a high-power output of 95.7 mW/cm^2^ and an OCV of 0.34 V were achieved, but the initial power output decreased to 82.5 mW/cm^2^ after only 20 min of operation because of the deterioration of the electrodes. 

To further increase the power-generating performance of NBFCs, not only single or binary noble metallic glucose-based NBFCs, but also ternary noble metallic NBFCs have been demonstrated [58,62]. Basu et al. [58] compared the power generation performances between bimetallic PdPt/carbon and tri-metallic PdPtAu/carbon, employed as GOx-like anode catalysts. The results indicated that the ternary metallic anode yielded a higher OCV but a similar power output compared to the bimetallic anode. Another interesting NBFC consisted of a thick film-type anode and cathode, manufactured by the e-beam evaporation technique, where the anode and cathode were made of PtNi alloy and porous Pd, respectively [63]. With a physiological glucose concentration (3.7 mM), the NBFC yielded a power output of 2.83 μW/cm^2^, an OCV of 0.35 V, and a current density of 8.2 μA/cm^2^. 

Metal-based nanozymes have been frequently anchored on carbon-based materials, particularly carbon nanotubes and graphene oxide (GO), to construct the electrodes of NBFC, owing to their advantageous characteristics as a support material such as excellent conductivity, large surface area, biocompatibility, insignificant mass, and possibility to tailor the catalytic activity and specificity toward a certain substrate. For example, Irfan et al. [64] designed a glucose biofuel cell including an anodic electrode made from a Ni/Co composite anchored on reduced GO (rGO), with a cathodic electrode made from Cu_2_O. Compared with Ni-rGO and Co-rGO, Ni/Co-rGO significantly improved the catalytic activity of glucose oxidation because of the synergistic effect of Ni, Co, and rGO. In particular, rGO served as a dispersant to prevent the aggregation of the Ni-Co composite, as well as a facilitator for the electron transfer between Co(III)/Co(II) and Ni(III)/Ni(II) couples when the glucose molecules attach to its grain boundary and sharp edges. The system showed a maximum power output of 28.807 W/m^2^, which was twice that of the fuel cell with a bare activated carbon electrode system. Another NBFC, composed of Pt/rGO as an anode and FeCo/Ketjen Black (KB) with 10 wt% polytetrafluoroethylene (PTFE) as a cathode, has been reported (Figure 4) [53]. With the level of glucose fuel (5 mM) and flowrate (0.33 mL/s) controlled to the same as those present in vein blood in a human arm, the NBFC yielded high potential of 388 mV of OCV at the first cycle, which could effectively be maintained with marginal power loss by successful circulation to remove the intermediate molecules. As this system utilizes circulating glucose as in the human vein, it could be utilized as a power source within an implantable device. Su et al. [65] also reported an interesting glucose-based NBFC, which included graphene sheets grafted with Pt and Pd as the anode and nitrogen-doped GO nanoribbons as the cathode. With physiological glucose (4 mM) in neutral cerebrospinal fluid, a high OCV of 0.216 V and maximal power output of 8.96 µW/cm^2^ were obtained with great stability and durability, which maintained the initial power output during one week with only a 7.9% decrease. The energy generation of this NBFC was more efficient at high temperatures, yielding a higher maximal power output of 24.9 µW/cm^2^ at 80 °C, demonstrating its potential for use in extreme weather environments. 

Noble metal-based NBFCs were also extended to construct implantable devices to generate electricity from real biological fluids. NBFCs composed of buckypaper modified with Au_80_Pt_20_ nanoparticles and carbon paper modified with Pt nanoparticles as anode and cathode, respectively, were implanted in orange pulp [66]. The implanted NBFCs successfully utilized glucose and fructose within the orange, and a high OCV (0.36 V) and power output of 182 μW were produced. The generated power was successfully applied to wireless signal transmission and the initial power was preserved for 7 h. Implantable NBFCs on humans have also been developing. For example, NBFCs composed of Au/carbon black and Au_60_Pt_40_/carbon black as anode and cathode, respectively, were demonstrated to produce a power output of 104 μW with human blood serum at normal glucose levels (5.4 mM) [67]. The power produced from the NBFCs was amplified with an energy harvesting circuit and successfully applied to a pace-maker over 10 h, showing the promising potential of NBFC-based implantable devices.

Overall, noble metal-based nanozymes have been intensively applied to construct an anode to oxidize glucose as well as a cathode to accept electrons in glucose-based NBFCs, which show a promising level of power generation. Nevertheless, their shortcomings, such as high cost and toxicity, have hampered their practical applications. In addition, when glucose-based NBFCs were operated with the real physiological fluids, many critical problems could happen, including biofouling, interfering effects from diverse biological substances, relatively low glucose level, and limited reaction environments, all of which reduce the biofuel cell performance and durability. Hence, the development of facile synthesis at low cost with engineered high biocompatibility and substrate specificity is required for widespread utilization. 

### 3.2. Metal Oxide-Based Nanozymes for Glucose-Based NBFCs

The high synthesis cost of noble metal-type nanozymes can be surpassed by metal oxide-type nanozymes, which can be easily synthesized at a low cost. An interesting metal oxide-based NBFC was presented by Ho et al. [68], where CoMn_2_O_4_/carbon was employed as a GOx-like anode catalyst in a biofuel cell system. The NBFC showed a power output of 2.372 mW/cm^2^, which is comparable to that of a commercial Pt/carbon-based biofuel cell system. Another NBFC utilized Co_3_O_4_ grown on graphene oxide as a GOx-like nanozyme to prepare the anode, and N and Fe co-doped biowaste-derived activated carbon to prepare the cathode [52]. The biofuel cell showed a maximum power output of 12.81 μW/cm^2^ and OCV of 0.442 V, with 10 mM glucose. Interestingly, as the glucose concentration increased to 10 mM, the power output increased gradually and dramatically dropped at approximately 30 mM. This was ascribed to the effect of high glucose concentration, which might prevent the hydroxyl radical from approaching the electrode anode. According to the study by Slaughter et al. [69], ZnO nanosol was deposited on an Al/Au substrate to construct an anode, and Pt was used to prepare the cathode. This NBFC showed a power output of 16.2 μW/cm^2^ with an OCV of 0.84 V. The electricity-generating mechanism from glucose could be interpreted by the unique electronic structure of the anode, where Zn and O within ZnO served as the electron acceptor and donor, respectively, owing to the special valence band of ZnO. 

Likewise, although the glucose oxidizing activity of metal oxide-based nanozymes is relatively lower than that of noble metal-based nanozymes, there have been many approaches for their unique utilization in the construction of glucose-based NBFCs.

## 4. Conclusions, Current Challenges, and Prospects

Recently, enzyme-mimicking nanozymes have attracted intense interest as promising alternatives to natural enzymes. In particular, diverse types of noble metal and metal oxide-based nanozymes can mimic natural GOx and catalase to efficiently oxidize glucose without accumulating harmful H_2_O_2_, as well as laccase to accept electrons, both of which are essential for constructing the anode and cathode, respectively, in glucose biofuel cells. Compared with EBFCs, NBFCs have distinct advantages such as high operational stability and robustness with extended lifetime, sufficient power output by high catalytic activity for glucose oxidation, low production cost by facile large-scale synthesis and manufacturing, and further tailored functionalities derived from the uniqueness of nanozymes. In this regard, the use of nanozymes in the glucose biofuel cell system significantly enhanced power generation performance (Table 1). 

Nowadays, wearable or implantable technology is becoming increasingly important since it allows people to track their body conditions as well as manage them, particularly through monitoring health risk factors. To this, efficient and sustainable power supply is highly necessary, and thus, glucose-based NBFCs have a crucial importance. For extensive practical applications of NBFCs, we believe the following technical issues should be addressed: (1) Until now, relatively limited types of nanozymes, primarily noble metal-based ones, have been utilized in NBFCs. Therefore, further advancement in the development of novel nanozymes composed of other elemental compositions or structures to possess higher catalytic activity and specificity toward target glucose would be highly desirable. Notably, the recent development of novel nanozymes includes many unique strategies, including single-atom nanozymes yielding significantly enhanced catalytic activity [70], active site-resembling nanozyme yielding both high activity and substrate specificity [25], and molecule-imprinted nanozyme yielding enhanced substrate specificity [71]. The rational design of nanozymes through the calculation of Gibbs free energy during the target catalytic reaction by density functional theory has also been introduced [72]. With these new and advanced methodologies to synthesize nanozymes, advanced NBFCs can be realized. (2) For practical usage of NBFCs to generate electricity from physiological glucose, the biocompatibility or safety of NBFC should be confirmed, particularly for devices implantable in the human body. (3) In addition, it is important to efficiently control electron transfer within NBFCs, which is an important factor in determining the power generating performance of NBFCs. With the above-mentioned studies, we expect that the utilization of NBFC will increase in the near future. 

## Figures and Tables

**Figure 1 nanomaterials-11-02116-f001:**
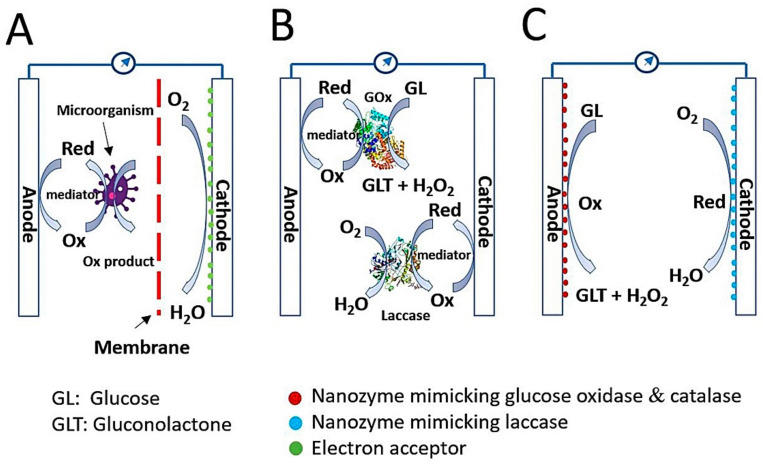
Three representative types of glucose biofuel cells. (**A**) Microbial biofuel cell (MBFC) architecture consists of an anode and cathode constructed using microorganisms, mediator, and supportive electrodes. (**B**) Enzymatic biofuel cell (EBFC) architecture includes an anode and cathode constructed using natural enzymes and mediator. (**C**) Nanozyme-based biofuel cell (NBFC) architecture includes an anode and cathode constructed using nanozymes.

**Figure 2 nanomaterials-11-02116-f002:**
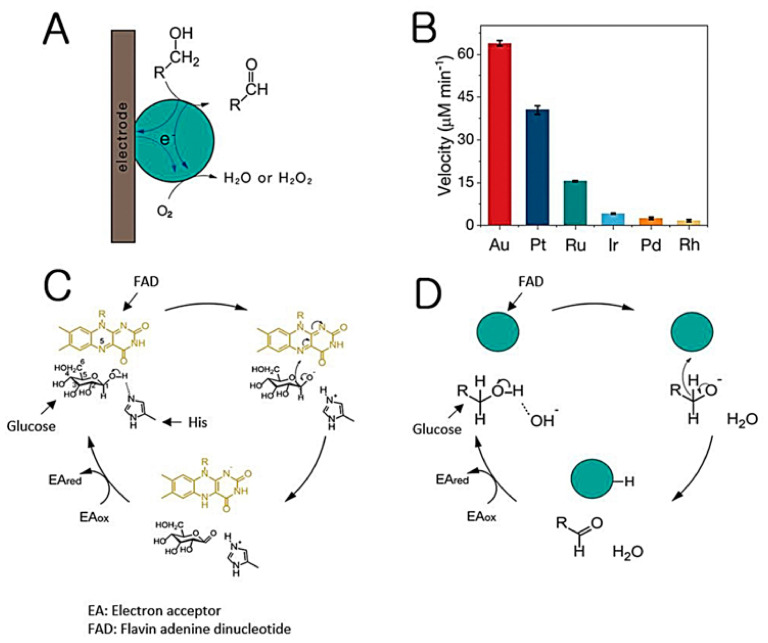
(**A**) Schematic illustration of electron transfer in the electrocatalytic oxidation of alcohols and the oxygen reduction reaction. (**B**) Comparison of the reaction rates. (**C**) Mechanism of glucose oxidation catalyzed by natural glucose oxidase (GOx). (**D**) Mechanism of glucose oxidation catalyzed by noble metal nanozyme. Reprinted with permission from ref. [35]. Copyright 2021 Springer Nature.

**Figure 3 nanomaterials-11-02116-f003:**
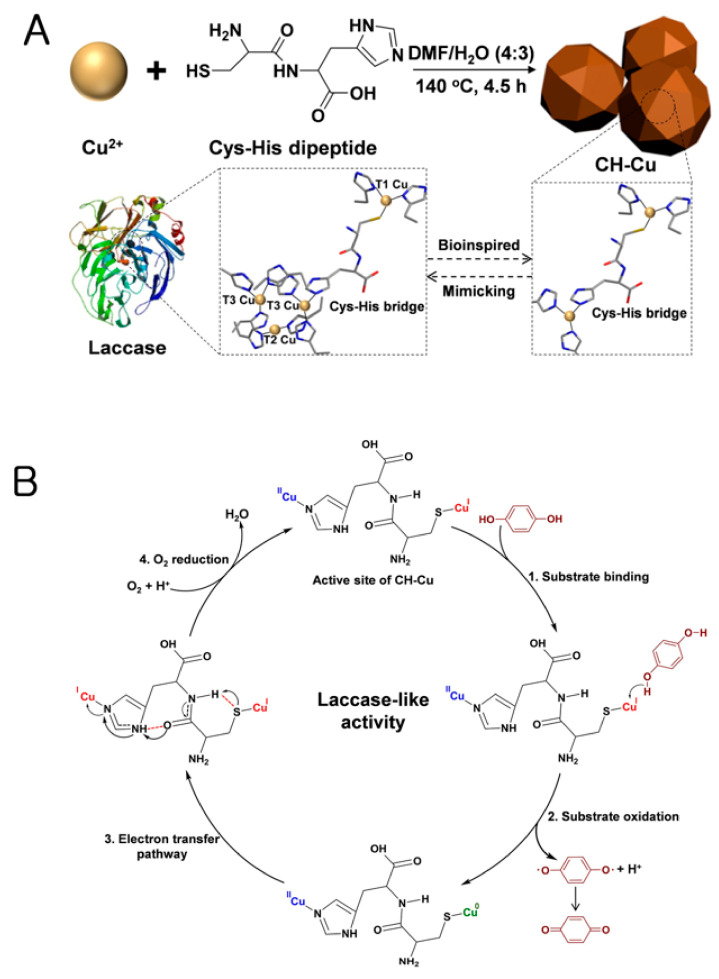
(**A**) Schematic illustration of the construction of laccase mimicking Cys-His (CH)-Cu nanozymes, which preserve many catalytic centers mimicking the active site of natural laccase. (**B**) Schematic illustration of possible catalytic mechanism involving the CH-Cu nanozymes. Reprinted with permission from ref. [47]. Copyright 2019 Elsevier.

**Figure 4 nanomaterials-11-02116-f004:**
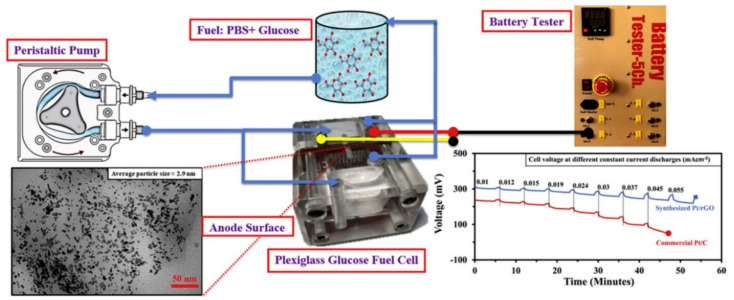
Schematic description of the nanozyme-based glucose biofuel cell system showing the potential to be utilized as a power source within implantable devices. Reprinted with permission from ref. [53]. Copyright 2020 Elsevier.

**Table 1 nanomaterials-11-02116-t001:** Diverse studies on nanozyme-based glucose biofuel cell.

Nanozymes	Anode	Cathode	Lifetime	OCV (V)	*P_max_* (mW/cm^2^)	Ref
Au film	Au film	Pt	90% retention after 60-day storage	0.916	0.307	[54]
Au nanowires	Au nanowires	Pt/carbon	93% retention after 30-day storage	0.425	0.126	[55]
Nano/micro hybrid structured Au, graphene	Nano/micro hybrid structured Au	Graphene	85% retention after 90-day storage	8.2	10.7	[56]
Pt/Au nano-alloy, graphene	Pt/Au	Graphene	NA ^i^	0.42	0.32	[61]
PtBi decorated nanoporous gold and Pt/carbon	PtBi decorated nanoporous gold	Pt/carbon	NA	0.95	8	[50]
Au_80_Pt_20_ nanoparticles/carbon black and Pt/carbon	Au_80_Pt_20_ nanoparticles/carbon	Pt/carbon	NA	0.34	95.7	[51]
PdPtAu/carbon, PdPt/carbon and Pt	PdPtAu/carbon, PdPt/carbon	Pt	NA	0.92	0.52	[58]
PtNi alloy and Pd	PtNi alloy	Pd	NA	~0.35	~0.00283	[63]
Ni foam and CoMn_2_O_4_/NC nanocomposites ^ii^	Ni foam	CoMn_2_O_4_/NC air cathode	~80% retention after ~7 h running	0.77	2.372	[68]
Graphene-cobalt oxide nanocomposite on Ni foam substrate and N, Fe-codoped biomass carbon	Graphene-cobalt oxide nanocomposite on Ni foam substrate	N, Fe-codoped biomass carbon	~80% retention after 10 h running	0.442	0.01281	[52]
Bimetallic Ni-Co composite anchored on reduced graphene oxide and Cu_2_O	Bimetallic Ni-Co composite anchored on reduced graphene oxide	Cu_2_O-Cu	NA	NA	2.8807	[64]
Pt/rGO and FeCo/Ketjen Black	Pt/rGO	Fe-Co/Ketjen Black	~77% retention after 15 h running	0.388	NA	[53]
Pt and Pd graphene and nitrogen doped graphene oxide nanoribbons	Pt and Pd graphene	N-doped GO nanoribbons	~92% retention after 7-day storage	0.216	0.0249	[65]
ZnO seed deposited on the Al/Au and single rod Pt	ZnO seed deposited on the Al/Au	Single rod Pt	100% retention for 9 h running	0.840	0.0162	[69]

^i^ NA: not available; ^ii^ NC: Nitrogen-doped carbon.

## Data Availability

Not applicable.

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
