# Peer review of "Research Progress and Prospects of Nanozyme-Based Glucose Biofuel Cells"

_nanomaterials, 2021, doi:10.3390/nano11082116_

Round 1

Reviewer 1 Report

The review article entitled "Research Progress and Prospects of Nanozyme-Based Glucose Biofuel Cells" by M.Kim et al is well organized and written. Ergo I recommend acceptance of the review paper after the authors revise the manuscript based on my following remarks/suggestions that will enable further improvement of the article.

  1. Why catalysts just made of metals/metal oxides/other inorganic components is called by the term nanozymes? This question has to be answered clearly in the introduction section of the manuscript.
  2. Comment on the structure of nanozymes that allows them to mimic the activity of enzymes.
  3. Comment on the potential and economic viability of nanonzyme based glucose biofuel cells for large-scale manufacturing in the conclusion section.
  4. The authors should include few sentences explaining why they think nanozyme based glucose biofuel cell research is important.

Reviewer 3 Report

The authors have written a well-focused and timely review on the use of nanozymes in glucose biofuel cells. The coverage of the relevant references seems very good and the review is well organized with a useful table of systems discussed at the end.

In section 2.1, the authors describe top down versus bottom up approaches to preparation of the nanozymes. It would be helpful if they could comment there on which upcoming studies are top down approach and which are bottom up approach and overall which strategy is yielding the better results for these biofuel cells and which one is more commonly used.

Additionally, can any comment be made on the preferred nanoparticle size range for good nanozyme activity?
